# The Impact of Whole Genome Data on Therapeutic Decision-Making in Metastatic Prostate Cancer: A Retrospective Analysis

**DOI:** 10.3390/cancers12051178

**Published:** 2020-05-07

**Authors:** Megan Crumbaker, Eva K. F. Chan, Tingting Gong, Niall Corcoran, Weerachai Jaratlerdsiri, Ruth J. Lyons, Anne-Maree Haynes, Anna A. Kulidjian, Anton M. F. Kalsbeek, Desiree C. Petersen, Phillip D. Stricker, Christina A. M. Jamieson, Peter I. Croucher, Christopher M. Hovens, Anthony M. Joshua, Vanessa M. Hayes

**Affiliations:** 1Garvan Institute of Medical Research, Darlinghurst, NSW 2010, Australia; m.crumbaker@garvan.org.au (M.C.); e.chan@garvan.org.au (E.K.F.C.); t.gong@garvan.org.au (T.G.); w.Jaratlerdsiri@garvan.org.au (W.J.); R.Lyons@garvan.org.au (R.J.L.); a.Haynes@garvan.org.au (A.-M.H.); amfkalsbeek@gmail.com (A.M.F.K.); p.croucher@garvan.org.au (P.I.C.); 2St. Vincent’s Clinical School, University of New South Wales, Sydney, Randwick, NSW 2031, Australia; 3Kinghorn Cancer Centre, Department of Medical Oncology, St. Vincent’s Hospital, Darlinghurst, NSW 2010, Australia; 4Central Clinical School, University of Sydney, Sydney, Camperdown, NSW 2050, Australia; 5Australian Prostate Cancer Research Centre Epworth, Richmond, VIC 3121, Australia; niallmcorcoran@gmail.com; 6Department of Surgery, University of Melbourne, Melbourne, VIC 3010, Australia; 7Division of Urology, Royal Melbourne Hospital, Melbourne, VIC 3050, Australia; 8Department of Orthopedic Surgery, Scripps Clinic, La Jolla, CA 92037, USA.; Kulidjian.Anna@scrippshealth.org; 9Orthopedic Oncology Program, Scripps MD Anderson Cancer Center, La Jolla, CA 92037, USA; 10The Centre for Proteomic and Genomic Research, Cape Town 7925, South Africa; desiree.petersen@cpgr.org.za; 11Department of Urology, St. Vincent’s Hospital, Darlinghurst, NSW 2010, Australia; phillip@stricker.com.au; 12Department of Urology, Moores Cancer Center, University of California, San Diego, La Jolla, CA 92037, USA; camjamieson@health.ucsd.edu; 13School of Biotechnology and Biomolecular Sciences, University of New South Wales, Sydney, Randwick, NSW 2031, Australia

**Keywords:** prostate cancer, precision medicine, whole genome sequencing, optical mapping, therapy

## Abstract

*Background*: While critical insights have been gained from evaluating the genomic landscape of metastatic prostate cancer, utilizing this information to inform personalized treatment is in its infancy. We performed a retrospective pilot study to assess the current impact of precision medicine for locally advanced and metastatic prostate adenocarcinoma and evaluate how genomic data could be harnessed to individualize treatment. *Methods*: Deep whole genome-sequencing was performed on 16 tumour-blood pairs from 13 prostate cancer patients; whole genome optical mapping was performed in a subset of 9 patients to further identify large structural variants. Tumour samples were derived from prostate, lymph nodes, bone and brain. *Results*: Most samples had acquired genomic alterations in multiple therapeutically relevant pathways, including DNA damage response (11/13 cases), PI3K (7/13), MAPK (10/13) and Wnt (9/13). Five patients had somatic copy number losses in genes that may indicate sensitivity to immunotherapy (*LRP1B, CDK12, MLH1)* and one patient had germline and somatic *BRCA2* alterations. *Conclusions*: Most cases, whether primary or metastatic, harboured therapeutically relevant alterations, including those associated with PARP inhibitor sensitivity, immunotherapy sensitivity and resistance to androgen pathway targeting agents. The observed intra-patient heterogeneity and presence of genomic alterations in multiple growth pathways in individual cases suggests that a precision medicine model in prostate cancer needs to simultaneously incorporate multiple pathway-targeting agents. Our whole genome approach allowed for structural variant assessment in addition to the ability to rapidly reassess an individual’s molecular landscape as knowledge of relevant biomarkers evolve. This retrospective oncological assessment highlights the genomic complexity of prostate cancer and the potential impact of assessing genomic data for an individual at any stage of the disease.

## 1. Introduction

Worldwide, prostate cancer (PCa) is the most commonly diagnosed non-cutaneous cancer in men and a leading cause of cancer-related male deaths [1]. Treatment strategies range from observation alone to multi-modal treatment and vary based on clinical and pathological factors such as tumour stage (T Stage), prostate specific antigen (PSA) level, Gleason or International Society of Urological Pathology (ISUP) score and life expectancy. PCa is a heterogeneous disease, and these clinical factors alone cannot predict outcomes accurately. Early disease is potentially curable whereas eventual treatment resistance is an intractable problem in metastatic disease. In both early and advanced disease, escalation of treatment including combination therapies, such as androgen deprivation therapy (ADT) administered with docetaxel with or without radiotherapy to the primary has resulted in improved outcomes [2,3,4,5]. However, the escalated treatment comes at the cost of increased toxicity and only a subset of men garner benefit. As such, predictive biomarkers for optimal treatment selection are needed at all stages of the disease. 

PCa progression is driven by genomic alterations and, as such, large sequencing efforts have focused on elucidating the succession of events driving its pathogenesis and progression. These sequencing efforts aim to establish new prognostic and therapeutic targets [6,7,8,9,10,11]. Thus far, these studies have focused on primary tumours from localized cancers and/or heavily pre-treated disease that has become resistant to ADT, termed castrate-resistant prostate cancer (CRPC). It has been established that intra- and inter-patient heterogeneity is high [12,13,14,15,16], though certain critical events may occur early in some patients and propagate. In general, genomic changes are thought to accumulate in response to treatment as the disease progresses and the importance of structural variants (SVs) in advanced prostate cancer is an evolving area of research [10,11].

A goal of previous studies is to advance “precision medicine” in PCa. Robinson et al. found that 89% of CRPC samples harboured a clinically actionable genomic alteration [6]. However, clinical trials utilizing the precision medicine paradigm of selecting a targeted drug based on molecular criteria have yielded mixed results. For example, though phosphoinositide 3-kinase (PI3K)/Protein Kinase B (Akt) pathway activating alterations are commonly reported in PCa, PI3K inhibitors have demonstrated limited efficacy to date [17,18]. However, a randomized phase II study of abiraterone +/−ipatasertib, an Akt inhibitor, in metastatic CRPC did find improved antitumoral activity in the combination arm, particularly in men with *PTEN* loss [19]. Similarly, poly ADP ribose polymerase (PARP) inhibitors have shown promise in selected men with CRPC and homologous recombination deficiency [20,21,22,23] but not all mutations in the homologous recombination pathway predict a response [21]. 

Although these large genomic studies have expanded the knowledge of molecular drivers of treatment-naïve primary and metastatic CRPC, they have generally viewed the data as a cohort without looking at cumulative alterations and their potential therapeutic impact within the individuals. Likewise, hormone sensitive (HSPC) metastatic disease has also been largely neglected. In this study, we performed whole genome sequencing (WGS) on men with confirmed PCa in order to assess the collective genomic events in individual cases and their impact on real-world therapeutic decisions. Recognizing the importance of SVs in prostate cancer and the limitations of WGS in detecting large genomic rearrangements, we also performed whole genome optical mapping (WGM) on a subset of the samples.

## 2. Results

### 2.1. Shared Genomic Landscape

In this study, we retrospectively analysed 13 PCa cases that had micro- or macro-metastatic disease at the time of sampling for genomic interrogation. Patient clinical and pathological characteristics are summarized in Table 1. Sixteen tumour samples comprised of nine primary and seven metastatic biopsies and the sites of concurrent or subsequent metastases included: bone (seven cases), lymph nodes (four cases), and brain (one case), while a single case had biochemically relapsed without evidence of macro-metastatic disease on conventional imaging. Figure 1 summarizes the commonalities and differences in the genomic landscape between our primary and metastatic samples, while placing our cases in context with the current knowledge based on large PCa WGS efforts. For the latter, we focused on the study published by Wedge et al. in 2018 for 112 patients (92 primary and 20 metastatic) with the metastases evenly distributed between HSPC and CRPC and biased towards lymph node metastasis (15/20) [8]. 

Common predisposing germline variants (Appendix A) in our samples include the *EHBP1* rs721048 (c.1185 + 30064G > A) intronic variant in five (38%) and *FGFR4* rs35185519011 (c.1162G > A, p.Gly388Arg) in ten (77%) cases. Reported in 9% of the healthy population [24], the *EHBP1* rs721048 A-allele has been associated with a more aggressive PCa [25]. The functional variant in *FGFR4*, although present in 30% of the healthy population, may predispose PCa patients to an accelerated disease course [26]. Ten patients had one of two SNPs (rs1859962, rs8072254) in non-coding regions of the 17q24.3 locus previously associated with PCa susceptibility [27].

Common somatic alterations include *ETS* fusions (seven cases) and *TP53* alterations (six cases). Tumour mutational burden (TMB) was generally low, ranging from 0.73 to 5.79 mutations/Megabase (mut/Mb) (IQR 1.30–2.09), and did not correlate with disease stage at sampling (Figure 1, Appendix A). Percent genome altered (PGA) ranged from 2.2% to 63.9% (IQR 2.79–19.4%) (Appendix A). 

We observed a prevalence (11/13 cases) of somatic copy number alterations (SCNA) affecting at least one DNA damage response (DDR) pathway gene (Appendix A). Losses in *FANCA*, which helps recruit DNA repair proteins to areas of DNA damage [28], were present in five (38%) cases, while one case harboured germline and somatic *BRCA2* alterations. With variation depending on the gene sets tested for and stage of disease, DDR gene alterations occur in approximately 20% of PCas with *BRCA2* alterations reported for 3% of prostatic and 12% of metastatic samples [6,7,29]. Aside from DDR, the most commonly impacted pathways were Phosphoinositide 3-kinase (PI3K, 7/13 (54%) cases), Mitogen-activated protein kinase (MAPK, 10 (77%) cases) and Wnt (9 (69%) cases) (Table 2). PI3K and MAPK are intracellular and extracellular signalling pathways, respectively, that are key to the regulation of the cell cycle and, like certain DDR pathways, are therapeutically targetable (manipulable) with inhibitory drugs [30,31]. The Wnt signalling pathway is a cellular pathway involved in cell growth, embryogenesis and cell cycle progression, the activation of which has been implicated in progression to CRPC and treatment resistance [32]. Previous studies have found that approximately 25% of primary PCas harbour PI3K or MAPK pathway alterations while nearly 50% of metastatic CRPC samples have PI3K alterations [6,7] and 32% MAPK amplifications [30]. In our study, 6 of the 7 samples with somatic alterations impacting PI3K were in the primary tissue, and MAPK alterations were seen in 4/7 (57%) of the metastases and 6/8 (75%) of the primaries. 

Overall, SCNAs and SVs, rather than single nucleotide variants (SNVs) and small insertions and deletions (sequences of no more than 50 nucleotides in length, indels), were more commonly acquired in PCa relevant genes (Appendix A). The addition of WGM identified 120 SVs not identified by WGS alone, several of which overlapped with oncogenic and/or tumour suppresser genes (Table 3, Appendix A). In particular, large insertions and duplications were typically missed by our short-read WGS approach. However, no recurrent WGM-derived SVs were observed across the cases. 

Recurrent non-coding events in key PCa-associated genes have been reported [8,11,33,34,35,36,37], including transcription factor (TF) binding sites. Alterations at key non-coding sites within our cases are summarised in Table 4. Common to CRPC, SCNAs or SVs upstream of the androgen receptor (AR) gene were not seen in our cases, which is unsurprising given the hormone sensitive status of most of our patients. All but two of our samples contained non-coding AR binding site mutations (Table 4). Overall, 20% of the somatic SNVs or indels affected at least one TF binding cluster. However, no sample was significantly enriched for mutations within TF binding clusters and no TFs were enriched for mutations. Notably, 0.3% of the 10.5 million TF binding clusters analysed correspond to *JUN*, an average of 1.2% (0.5–1.5%) of somatic SNV in *JUN* binding clusters. JUN is a transcription factor that antagonizes AR signalling [38]. 

Excluding COSMIC Mutational Signature 1 common to all cancers, we observed a predominance of Mutational Signatures 3 and/or 8 (Figure 2A) that generally reduced in proportion from the clonal to subclonal stages of tumour evolution (Figure 2B). Known to be associated with DDR gene alterations [39,40,41], Signature 8 was particularly common in the primary 8/9 (89%) versus metastatic 3/7 (43%) samples, with notable loss in both of case 19651’s lymph node metastases. In contrast, Signature 5, which is seen in most cancer types, particularly in smokers [39], and Signature 16, most often associated with liver cancer, both increased in the subclonal stage of tumour evolution. 

When viewed in detail, each patient had unique features with potential therapeutic implications. This highlights the relevance of genomic information for guiding therapeutic decisions, including data derived from primary tumour tissue. Here, we discuss how the course of treatment for each patient may have been influenced by the availability of their genomic data.

### 2.2. Primary Prostate Samples with Synchronous Lymph Node Metastases: 19011, 19260, 19145 and 19651

These cases each presented with elevated PSA levels and prostate adenocarcinomas confirmed on biopsy. Only 19651 had evidence of nodal metastases on conventional imaging preoperatively. However, at radical prostatectomy with lymph node dissection, all had pathologically involved nodes. Genomic alterations of potential relevance are summarized in Figure 3 (19011, 19260, 19145) and Figure 4 (19651). Though nodal involvement at presentation is associated with a high PCa mortality rate [42], the optimal management strategy for these men has not been established. Retrospective data suggest that adjuvant ADT with radiotherapy compared to ADT or observation is beneficial for men with lymph node metastases identified at radical prostatectomy [43]. Based on their ISUP grade group or Gleason score and tumour (T) stage, 19011, 19145 and 19651 also meet eligibility criteria for the STAMPEDE trial arms C and G that have shown benefit for adding docetaxel or abiraterone respectively to ADT [3,44,45]. Though some studies of men with high-risk localized PCa treated with neoadjuvant and/or adjuvant docetaxel demonstrate improved outcomes, these improvements occur in a small proportion, with significant toxicity to many [4,46,47]. These studies have all been based on clinical risk factors, thus, there is an urgent need for biomarkers that better select men likely to benefit, thereby avoiding over- and undertreatment. 

#### 2.2.1. Case 19011: Left Prostate Tumour Core Biopsy

This *TMPRSS2-ERG* positive case had a somatic missense mutation in *MED12* (c.3670C > G; p.Leu1224Val), that is potentially pathogenic in many cancers, including prostate [48], via its upregulation of Wnt/β-catenin signalling [49]. A non-coding somatic SNV upstream of *CTNNB1* that is within the binding region of 188 TFs, may indicate misregulation of this gene involved in Wnt/β-catenin signalling [50]. No therapeutically relevant SCNA or SV was identified. PGA was low (2.7%). 

Although no targetable alteration is seen in this case, the lack of mutations and low PGA may still be valuable in guiding decision-making. This patient is unlikely to respond to targeted therapies, like PARP inhibitors, but also to non-targeted agents that rely upon high mutational loads, such as immune checkpoint inhibition. The lack of poor prognostic markers, such as *TP53* loss, could mean this low volume, locally advanced PCa may respond well to aggressive local therapy without escalation to systemic therapy (e.g., addition of docetaxel). A low PGA is associated with a lower risk of BCR following definitive local therapy [51]. Despite meeting criteria for perioperative therapy trials, his genomic profile suggests aggressive local therapy will be sufficient. However, should he relapse however, the alterations in Wnt pathway-associated regions could confer resistance to AR targeting agents [32].

#### 2.2.2. Case 19260: Right Prostate Tumour Core Biopsy

Patient 19260 was also treatment-naïve at the time of his prostatectomy and sampling for WGS. He biochemically relapsed 16 months postoperatively at which time he had salvage radiotherapy with a good PSA response.

*TMPRSS2-ERG* fusion positive with a low PGA (2.4%), this case presented with a pathogenic somatic missense mutation in *BRAF* (c1406G > T; p.Gly469Val). Known to confer increased kinase activity [48], this mutation may sensitise the patient to BRAF +/− MEK inhibitor therapy. Of interest in CRPC [30], with a report of response to targeted therapy in a *BRAF* mutant patient [52], clinical trials of MEK inhibitors are currently underway (NCT02881242). Though not relevant to this patient’s upfront treatment, it could prove useful in the event of relapse. 

#### 2.2.3. Case 19145: Left Prostate Tumour Core Biopsy

This *TMPRSS2-ERG* positive tumour had a high PGA (10.2%), but lacked any known deleterious somatic mutation. SCNAs/SVs of note include heterozygous losses in *PTEN, FANCA, CDK12, TP53, NCOR1* and *NCOR2*, an inter-chromosomal translocation with breakpoints overlapping *RAD51B* and *PTEN* and a large heterozygous deletion overlapping with *TP53* and *NCOR1*. 

Responses to PARP inhibition have been seen in patients with *FANCA* alterations [23,53] and preclinical data suggest that *PTEN* loss sensitises cancers to PARP inhibitors, with reported cases of exceptional responses to olaparib [54,55]. However, resistance to single agent PARP inhibition has been described in Pten/p53 deficient mouse models, though a synergistic response was seen upon PARP inhibition in combination with PI3K inhibition. [56]. *NCOR1* and *NCOR2* are transcriptional corepressors that negatively regulate androgen receptor (AR) signalling and androgen-induced cell proliferation [57,58,59]; losses in these genes increase with disease progression and are associated with anti-androgen and ADT resistance [60,61]. *TP53* loss may also predict inferior responses to novel androgen signalling inhibitors (ASIs), such as enzalutamide and abiraterone, in CRPC [62]. CDK12 loss may predict sensitivity to immune checkpoint inhibiting therapies [63].

Many of the observed alterations in this case have therapeutic potential but are still the subject of early phase clinical trials. The presence of the *NCOR1/2* losses, however, may indicate a vulnerability in this patient for early development of CRPC. His four week course of ADT preoperatively may have induced treatment resistant clones even at this early stage. These losses together with *TP53* loss and high PGA indicate this patient may develop early resistance to ADT and, given his high-risk disease at presentation, he would be an ideal candidate for escalation of his initial treatment with chemotherapy. 

#### 2.2.4. Case 19651: Bilateral Prostate and Internal Iliac Node Tumour Core Biopsies

Reporting a family history of PCa, via his father, and breast cancer in his mother and sister, it was not surprising that this patient carries a pathogenic germline *BRCA2* stop-gain mutation (rs80359031; c.7988A > T; p.Glu2663Val) confirmed to predispose carriers to BRCA-associated cancers. 

The somatic heterogeneity across the four tumour samples is striking (Figure 1 and Figure 4A). Of the 78 overlapping SNVs (out of 24,195) present across all four samples, none had notable therapeutic relevance. Phylogenetic reconstruction of this cancer’s evolution reveals distinct differences between the left primary and the other three samples (Figure 4B). Notably, the left prostatic primary acquired a somatic pathogenic *BRCA2* stop-gain mutation ((c.6308C > G; p.Ser2103Ter), variant allele frequency (VAF; 26%). Additionally, genes associated with several different growth signalling pathways, including MAPK/ERK, TGF-β, PI3K and WNT, are impacted by SCNAs in the left primary but there are few events in the other samples. No relevant SVs within the left lymph node were noted on WGM. As expected with the combined germline and somatic *BRCA2* mutations, there was a high rate of large deletions in the left primary [10], including a 3Mb deletion overlapping multiple tumour suppressor genes (TSGs) including *BTG* and *DCN*. 

Inter- and intra-patient heterogeneity have been well-described in PCa [13] and most recently in multi-focal primary tumours [64], with significant therapeutic implications. The germline mutation not only informs screening for secondary cancers and testing in relatives, *BRCA2* mutations may also be associated with a worse prognosis [65,66,67,68,69] and confer sensitivity to platinum-based chemotherapy [70] and PARP-inhibitors [23,53]. However, there is increasing evidence that responses are markedly improved with biallelic loss and many of the PARP inhibitor clinical trials have refined their inclusion criteria to include only patients with biallelic alterations. Acquiring a somatic *BRCA2* mutation in a single primary tumour could result in a differential response to targeted therapies that would not be predicted based on the typical single site sampling performed in clinical practice. 

Aside from the germline *BRCA2* mutation, there is no unifying therapeutically relevant event across all four samples. Having had short-term ADT preoperatively, losses in *NCOR1* and *NCOR2* as well as other SCNAs associated with CRPC within the left primary raise the possibility that early ADT resistance is developing after minimal treatment.

In practice, knowledge of this patient’s genomic landscape at baseline may have prompted his treating clinician to escalate his treatment with combination systemic therapy such as the rucaparib arm of the STAMPEDE trial. The loss of *NCOR1* and *NCOR2* and the poorer prognosis conferred by his *TP53* and *BRCA2* status represent potential indications for early chemohormonal therapy (ADT with docetaxel chemotherapy) despite him having low-volume, node only metastases [3,4,46,71]. *BRCA2* alterations may also sensitize this patient to radiotherapy due to impaired DDR. Therefore, had his genomic data been available early, an upfront strategy with radiotherapy to his primary in combination with ADT and docetaxel may have been used. At progression, he may be considered for a clinical trial with a PARP inhibitor, potentially in combination with another agent given his somatic *BRCA2* discordance. A metastatic biopsy at a site of progression could prove useful in determining whether new sites of disease harbour the somatic *BRCA2* alteration. 

### 2.3. Primary Prostate Samples with Relapse Post Radical Prostatectomy: 12543, 5545, 5684, and 13179 

At the time of surgery, none of these cases had evidence of metastatic disease on staging scans. All men subsequently relapsed with incurable disease, including bone metastases (5545, 5684 and 13179) and persistent BCR with eventual CRPC (12543). While TMBs were similar (range 1.4–1.9), there was more marked variability in their PGAs (range 3.1–26%). Genomic alterations with patient-specific relevance are summarized in Figure 5. 

#### 2.3.1. Case 12543: Left Prostate Tumour Core Biopsy

This patient’s tumour is characterized by *KMT2C* mutation, copy number losses (supported by large deletion) in *PTEN* and *FOXP1* and an *ETV1-ACSL3* fusion. *ETV1-ACSL3* fusion may account for this patient’s prolonged ADT sensitivity (no evidence of metastatic disease following 10 years on ADT for BCR). *ACSL3* is an androgen responsive gene and thus, this fusion may lead to a strong reliance on androgen signalling [72]. Despite *PTEN* loss, loss of *FOXP1* may restore androgen receptor signalling, further enhancing this patient’s response to ADT despite the *PTEN* loss [73]. At development of CRPC, this reliance on AR signalling may be exploited further with the addition of a novel ASI to his ADT, rather than docetaxel.

#### 2.3.2. Case 5545: Left Prostate Tumour Core Biopsy

This case is characterized by a deleterious somatic *SPOP* missense variant (rs193921065, c.399C > G; p.Phe133Leu; VAF 44%) [7] and a large hemizygous deletion encompassing *CHD1*. We also predict the *LRB1B* mutation (c.3178A > G; p.Cys1060Arg) to be deleterious. Notable copy number losses include *TP53BP1* and the TSG *RB1* and the DDR genes *FANCA* and *PPP2R2A*, while a deletion overlapped *LRP1B.* Unique to WGM, we identified a large deletion involving *FILIP1L*, a gene commonly hypermethylated in PCa [74].

Point mutations in *SPOP* occur in approximately 11% of primary PCas [7] and are commonly associated with *CHD1* loss [75]. This combination of alterations is associated with increased abiraterone sensitivity in CRPC [76]. These tumours are also characterized by increased genomic instability due to error-prone double-strand DNA break repair, which results in more SVs, as seen in this case, and potential vulnerability to DNA damaging treatment such as irradiation, PARP inhibition and platinum chemotherapy [77]. Loss of *FANCA,* a gene involved in homologous recombination, may also sensitise this cancer to PARP inhibition. A recent retrospective study found that *LRP1B* alterations may predict for sensitivity to pembrolizumab [78]. 

*SPOP/CHD1* co-altered clones persist across the disease spectrum in studies of serial patient samples [76]. Therefore, knowledge of this case’s genomic data from radical prostatectomy would lead to a preference for abiraterone over docetaxel at development of CRPC. These alterations may also increase his responsiveness to PARP inhibition, though evidence is limited and preclinical models have shown that this vulnerability is reliant on elevated 53BP1 protein levels [77] and so the copy number loss in *TP53BP1* may counteract this vulnerability. This combination of alterations highlights the importance of understanding the entire genomic landscape in an individual. 

#### 2.3.3. Case 5684: Right Prostate Tumour Core Biopsy

This case harbours a small frameshift deletion in *TP53* between exons 11 and 12, in addition to heterozygous copy number loss and a deletion on SV analysis. He also presented with SCNA in *CDH1* and alterations in other DDR genes including: an SNV in *CDK12* and SCNAs in *PPP2R2A* and *FANCA.* Losses in genes affecting proliferative pathways include those in *PIK3R1,* the loss of which activates the PI3K pathway [79] and *MAP3K1,* which is associated with MEK signalling [80]. The inclusion of WGM for 5684 revealed a large insertion overlapping *SPOCK1*, which encodes a protein found to promote tumorigenesis and metastases in PCa [81]. WGM also identified an insertion in *CREBBP*, a coactivator of AR that is usually overexpressed in CRPC and the upregulation of which is associated with ADT resistance [82].

*TP53* loss confers a worse prognosis and improved outcomes with chemotherapy compared to novel ASI agents [62]. Knowledge of his primary tumour *TP53* status may have guided ordering of therapies with a preference for chemotherapy, particularly upon progression to CRPC. A study of co-targeting PARP and Wee1 kinase with olaparib and AZD1775 is currently underway for *TP53* mutated solid tumours (NCT02576444). The losses in *PIK3R1* and *MAP3K1* may confer sensitivity to PI3K and MEK inhibitors respectively though these agents would only be used on a suitable clinical trial. 

#### 2.3.4. Case 13179: Right Prostate Tumour Core Biopsy

This *TMPRSS2-ERG* positive tumour is characterised by a high PGA at 26%, pathogenic somatic SNV in *TP53* (rs28934575, c.733G > A; p.Gly245Ser), as well as copy number loss of *PTEN*. Additional losses in *MAP3K1, PIK3R1* and *TP53* were observed, along with somatic alterations in the MAPK and PI3K pathways (Figure 5). 

Co-loss of *TP53* and *PTEN* is associated with more aggressive disease, which is consistent with this patient’s clinical course. Knowledge of these molecular features may have triggered more aggressive treatment upfront. Within current treatment paradigms, this may have included radiotherapy with ADT and docetaxel [5,71]. Additionally, the number of alterations in multiple targetable pathways, particularly PI3K (PI3K/AKT inhibitors) and MAPK/ERK (BRAF/MEK inhibitors), highlights the need to contextualise genomic events rather than viewing them in isolation. It is likely that this patient’s treatment regimen would need to involve a tailored combination strategy if a targeted, precision-medicine approach was to be considered.

### 2.4. Bone Metastatic Samples: 147, A153, PCSD13 and 1135

Sampling for genomic analyses occurred at bone biopsy. Patients 147 and A153 had not yet had systemic therapy, while 1135 had CRPC, having commenced intermittent ADT for BCR 3 years postoperatively. PCSD13 presented with de novo metastatic disease manifesting as hip pain. Investigations revealed multiple bone metastases and an elevated PSA. Selected genomic events are summarized in Figure 6. 

#### 2.4.1. Case 147: Biopsy Left Pubic Bone Corresponding to Sclerotic Region on Imaging

This case did not have any relevant somatic SNVs or WGS-identified SVs. SCNAs included gains in *BRAF, AHNAK* and *BRD4*.

It is unknown, yet unlikely, whether the copy number gain in *BRAF* would be sufficient to sensitize the patient to BRAF inhibition. The low level of relevant alterations in this case may explain his less aggressive disease course with a late clinical relapse (10 years post prostatectomy). The gains in *BRD4* and *AHNAK* may have contributed to metastasis formation: BRD4, part of the Bromodomain and Extraterminal (BET) protein family, regulates tumour cell migration and invasion through transcription of *AHNAK* [83]. Small molecule BRD4-selective degraders inhibit metastatic potential in PCa cell lines and a Phase I clinical trial of birabresib which included CRPC patients has been completed [84]. BRD4 is also involved in the non-homologous end joining (NHEJ) DDR pathway and higher protein levels from pre-treatment biopsies are associated with poor outcomes following radical radiotherapy in localized disease [85].

#### 2.4.2. Case A153: Biopsy Right Iliac Crest Corresponding to Metastatic Deposit on Imaging

This *TMPRSS2-ERG* positive metastatic tumour harboured a pathogenic *TP53* mutation (rs121912656, c.734G > T; p.Gly245Val) and a high PGA (25.5%). SCNAs include losses in *APC, PTEN, CHD1, BRCA2, FANCA, PIK3R1* and *LRP1B*. A complex SV on chromosome 5 encompassing *PPAP2A, PDE4D, MAP3K1* and *IL6ST,* was previously associated with a worse prognosis [8]. 

*TP53* loss is associated with a worse prognosis and decreased response to abiraterone in CRPC. *APC* loss, through its activation of Wnt signalling, may promote ASI resistance [32,62]. These two features would make docetaxel a better option than an ASI in the first instance for this patient at metastatic relapse. *BRCA2* and *FANCA* alterations were predictive for sensitivity to olaparib in the TOPARP studies [23,53] and, as previously discussed, *PTEN* and *CDH1* losses may sensitize this patient to PARP inhibition [54,77]. 

#### 2.4.3. Case PCSD13: Biopsy Left Femur during Total Hip Replacement for Pathological Fracture

PCSD13 presented with a pathogenic germline *IDH2* mutation (rs121913502, c.419G > A; p.Arg140Gln). Reported to have an allele frequency of 0.00003 in The Genome Aggregation Database (gnomAD) [86], while associated with several other cancers, this mutation has not yet been reported in PCa [87]. In addition to an SNV in *AKT1*, there is a copy number gain in this gene. There are losses in the DDR genes *CDK12* and *MLH1*, and SVs also overlap multiple DDR genes. The COSMIC Mutational Signatures in this case show a subclonal increase in the proportion of Signature 3, whereas the majority of the other samples showed a decrease in this signature, which is associated with failure of double-strand DNA repair (Figure 2B).

The *AKT1* alterations may have contributed to his early ADT resistance (within 3 months of starting ADT) and confer sensitivity to AKT inhibitors [19]. These alterations could influence decisions on escalating ADT treatment with the addition of abiraterone, an androgen targeting drug, or docetaxel. However, the crosstalk between AR and PI3K/AKT signalling is well-established, [88,89] and additional pressure on the androgen axis in the context of an *AKT1* amplification may only drive further growth via the PI3K pathway. In the absence of a clinical trial with an AKT inhibitor, the addition of docetaxel rather than an AR targeting agent may have been more prudent. Immune checkpoint inhibition may have been another treatment option for this patient with his *CDK12* and *MLH1* SCNAs. This patient succumbed to his cancer shortly after developing CRPC.

#### 2.4.4. Case 1135: Biopsy Right Posterior Iliac Crest Corresponding to Metastatic Deposit on Imaging

Despite having CRPC at the time of biopsy, case 1135 had very few alterations of interest with a TMB of 0.73 and PGA of 3.1%. This tumour contained SNVs in *KMT2C* and *IDH2* (rs121913502, c.419G > A; p.Arg140Gln) and SCNAs in *BCOR, NCOA7,* and *NOTCH2*. No significant SVs were identified with WGS but a homozygous deletion overlapping *TNS3* was identified using WGM. 

The somatic SNV in *IDH2* is the same as the germline alteration seen in PCSD13 that has not been reported in PCa. It is unclear whether this mutation would drive the progression of this patient’s cancer and if IDH inhibitor therapy, used to treat IDH-mutant AML, would be relevant. Based on preclinical studies, *KMT2C* alterations may confer sensitivity to PARP inhibition via its effects on the epigenetic status and expression of DDR genes. However, alterations in *KMT2C* are frequent in PCa [7] and responses to PARP inhibition only occur in a small proportion of patients [23]; therefore, it is unlikely this SNV alone will be enough to predict sensitivity to PARP monotherapy. ATRX is a DDR pathway gene while *BCOR, NCOA7* and *NOTCH2* are involved in androgen signalling. However, these alterations do not yet have any targeted therapeutic strategies for CRPC. While the impact of the deletion in *TNS3* is again unclear, it is noted that Tensins are a family of scaffolding proteins that regulate cell motility and growth and *TNS3* in involved in MET signalling [90], a target of the tyrosine kinase inhibiting drug, cabozantinib. Overall, though this case’s alterations do not yet have any therapeutic relevance, the knowledge of molecular features in PCa is rapidly evolving and future findings may bring useful drugs to light.

### 2.5. Case 80002: Core Biopsy at Resection of Brain Metastasis

Patient 80002 presented with a solitary brain metastasis that was surgically resected. His PSA was elevated and morphology of the tumour specimen was consistent with an adenocarcinoma of prostatic origin; immunohistochemistry (IHC) markers for neuroendocrine differentiation were negative.

The relevant genomic features of this *TMPRSS2-ERG* fusion positive case are summarized in Figure 7 and include: *TP53* mutation (rs1057519999, c.716A > C; p.Asn239Thr) and SCNAs in *CDK12, RAD51C, RNF43, TP53,* and *BRAF*. This tumour presented with a high rate of SVs, including a large deletion overlapping *TP53,* a partial deletion of *LRP1B,* and an interchromosomal translocation involving *CTNNA1*, the downregulation of which is associated with a worse prognosis in PCa [91]. Using our WGM approach, we identified additional large heterozygous deletions. Two overlap TSGs including *TP53* and *KCTD11,* and another overlaps with *TBX3* [92] and *NRF2*. NRF2 has been shown to suppress PCa cell mitosis and migration [93,94]. Another large deletion on chromosome 2 overlapped *HOXD10* and *HOXD3*. Decreased HOXD10 expression promotes an aggressive phenotype in PCa in knockdown mice, as well on retrospective review of clinical outcomes [95] and *HOXD3* methylation predicts earlier BCR [96]. 

Although COSMIC Mutational Signatures 1, 5, 8 and 9 are present, Signatures 17 and 18 contribute >5% each. Signature 18 may be associated with failure of base excision repair [97] and enriched in metastatic PCa [8]. Signature 17, predominantly found in gastric and oesophageal cancers, has been shown to co-occur with Signature 18 in mouse models of these cancers and this signature may be a by-product of oxidative damage [98,99]. 

Brain metastases are uncommon in prostate adenocarcinoma and tend to occur in cases with neuroendocrine differentiation [100]. However, gains in *FOXA1,* as seen in this case, are thought to protect against neuroendocrine trans-differentiation [101] and the *TMPRSS2-ERG* fusion supports the prostatic origin. This patient has a number of targets impacting androgen signalling, DDR and MAPK pathways. His clinical presentation would already support aggressive therapy with combination therapy and his genomic data include several poor prognostic features. The partial *LRP1B* deletion may produce sensitivity to pembrolizumab but the evidence for this is limited so this should only be considered as part of a clinical trial potentially upfront with docetaxel or later in his clinical course at development of CRPC. *KCTD11* is a negative regulator of hedgehog pathway signalling [102] and therefore its loss, identified using WGM, may increase signalling and imply this tumour would be sensitive to pathway inhibitors. 

## 3. Materials and Methods 

Included cases had adenocarcinoma of prostatic origin and were selected based on availability of tissue and matched blood specimens and micro- or overt metastatic disease either at the time of sampling or subsequent to radical prostatectomy. Patients sampled at the time of radical prostatectomy (primary tissue) had either pathologically confirmed lymph node metastases (19011, 19145, 19260, 19651) at diagnosis or subsequent metastatic relapse post-surgery (5545, 5684, 12543, 13179). Patients recruited at presentation of distant metastases had bone (1135, 147, A153, PCSD13) or brain (80002) tissue sampled. 

All samples were obtained with written informed consent, as per the study approval granted from the St. Vincent’s Human Research Ethics Committee (HREC), SVH/12/231 and HREC/12/SVH/323, Melbourne Health Human Research Ethics Committee HREC/12/MH/272 and Epworth Health 55512, or University of California Institute Review Board (IRB) approval 090401. Samples were shipped to the Garvan Institute of Medical Research in accordance with institutional Material Transfer Agreements (MTAs), and genomic screening and analysis were performed in accordance with approval granted by St. Vincent’s Hospital HREC SVH/15/227 and governance review authorisation granted for human research at the Garvan Institute of Medical Research GHRP1522. 

Primary tumour samples were collected at the time of radical prostatectomy and two core biopsies were taken from the prostate regions with cancer on preoperative biopsy. Lymph node tissue was collected at the time of radical prostatectomy from nodal masses with palpable tumour. Metastatic samples were obtained by image guided biopsy or at surgical resection (80002, PCSD13). All tissue samples were snap frozen. The presence of prostate cancer and its location within the samples was confirmed by a pathologist prior to dissection for DNA extraction. DNA was extracted from tissue and buffy coat or whole blood using one of two commercially available kits: the DNeasy blood and tissue kit protocol (Qiagen, Maryland), or for high molecular weight (HMW) DNA, the Bionano Prep Frozen Human Blood and Animal Tissue DNA isolation protocols (Bionano Genomics, San Diego document #30246 and #30077). 

Demographic, clinical and pathological data were collected for each patient and are summarised in Figure 1 and Table 1. The median patient age at the time of PCa diagnosis was 65 years (range 51–77). The median time to biochemical recurrence (BCR) for those that underwent definitive first-line treatment and subsequently relapsed (*n* = 8) was 43.5 months (range 6–93); six of these patients relapsed with metastatic disease detectable on standard imaging at a median time of 86.5 months from initial diagnosis (range 24–120).

### 3.1. Whole Genome Sequencing (WGS) 

DNA from tumour and matched blood underwent 2 × 150 bp sequencing on an Illumina HiSeq X Ten instrument (Kinghorn Centre for Clinical Genomics, Garvan Institute of Medical Research) averaging over 80× and 40× coverage, respectively. Read adapters were trimmed using Illumina’s Bcl2fastq Conversion software (Illumina) and filtered to remove low quality bases (<Q15), short reads (<70 bp) and missing read pairs using cutadapt v1.9 [103]. Remaining reads were aligned to GRCh38 reference using bwa-mem v0.7.15 [104], with the ALT-aware mode. Alignment statistics were calculated using QualiMap v2.1.3 [105] and stromal contamination was calculated using Sequenza [106]. Sequencing statistics are summarized in Appendix A. The sequencing data for the tumor and blood samples are available in the NCBI BioSample database under the following range of accessions: SAMN14209964–SAMN14209992. 

### 3.2. WGS Variant Calling

The GATK pipeline version 3.5-0 was used for small variant calling [107]. We defined small variants as single nucleotide variants (SNVs) and insertions or deletions (indels) ≤ 50 bases and structural variants (SVs) as events ≥ 50 bases. Analysis-ready alignment per sample was called for SNVs and indels (GVCF mode) using GATK HaplotypeCaller (GVCF mode; [107]). Per-sample GVCFs were used for joint genotyping across genomes (GATK GenotypeGVCFs). Joint-called SNVs and indels were filtered via machine learning variant quality score recalibration and passed loci were kept. High-confidence somatic variants were called for each tumour-blood pair using MuTect2 [108]. A combination of GRIDSS and LUMPY was used for the detection of germline and somatic SVs [109,110]; potentially relevant SVs were manually inspected using Integrative Genomics Viewer (IGV) [111]. For somatic copy number alterations (SCNAs), binned copy number and segmentation profiles were determined using the copy number calling pipeline in the CNVkit package; gains (CN > 2) and losses (CN < 2) were assessed on calls adjusted for tumour purity [112]. 

### 3.3. WGS Variant Annotation

Germline and somatic SNVs and indels were annotated using Annovar [113] and pathogenic variants were manually inspected using IGV [111]. Missense mutations were further classified as potential oncogenic drivers using CanDrA [114] with PCa-specific databases.

The 30 SNV-derived Catalogue Of Somatic Mutations In Cancer (COSMIC) Mutational Signatures were annotated using the SomaticSignatures package in R [115]. Estimation of clonality and clonal segregation of somatic mutations were computed using PhyloWGS [116] and TITAN program [117].

### 3.4. Tumour Mutational Burden and Percentage Genome Alteration

TMB was calculated by counting the total number of small somatic mutations and dividing by genome size (3088 megabases (Mb)). PGA was calculated based on the cumulative number of base pairs altered for each gain or loss in the autosome (Chromosomes 1–22) per patient divided by the reference autosomal genome size (2875 Mb). 

### 3.5. Whole Genome Optical Mapping

HMW DNA were fluorescently-labelled using either nicking enzyme Nt.BspQI (New England Biolabs) or non-nicking enzyme DLE-1 (BNG, Part #20351), according to the Bionano Prep Labeling NLRS Protocol (Document #30024) or Direct Label & Stain protocol (Document #30206), respectively. Samples prepared with BspQI (1135, 147, A153) were imaged using the Bionano Genomics (BNG) Irys system (San Diego, CA), while those prepared with DLE-1 (80002, 19651, 12543, 13179, 5545, 5684) were imaged using the BNG Saphyr system, to generate single molecule optical maps. 

De novo assembly of single molecules into consensus genome maps was performed with the Bionano Solve (≥v3.2) software with aligner RefAligner (≥7437) [118,119,120]. Custom sets of parameters were used for this purpose and are included as File S1. 

### 3.6. WGM Derived Genomic Rearrangements

SVs were identified relative to the human reference genome, GRCh38, whose genome maps were bioinformatically deduced based on predicted Nt.BspQI (GCTCTTCN) or DLE-1 (CTTAAG) motif sites. SV detection was performed as part of the Bionano Solve pipeline. Details of the underlying algorithm are described in the software’s accompanying documentation (Document # 30110B). 

### 3.7. WGM Derived Data Filtering

Filtering steps were performed on the resulting SVs. First, SVs that did not pass the Bionano recommended confidence level for the corresponding SV type were excluded; that is, all SVs other than inversion must have confidence > 0.1 and both breakpoints of an inversion event must have confidence > 0.01. Second, only rare SVs were included, defined as being observed in < 10% of a cohort of ~150 “normal” samples provided by Bionano [121]. Finally, “somatic” SVs were identified as those supported by a minimum of *y_t_* molecules in the tumour sample but not observed in more than *y_n_* molecules in the matching-normal sample, where *y = −0.3 + 0.13 * x* and *x* being the effective coverage of the corresponding sample. This formula is recommended by Bionano as detailed in their Variant Annotation Pipeline v1.0 (BNG document # 30190). The minimum coverage cut-offs for somatic SV calling are summarized in Appendix A. The WGM data are available at the following Doi: 10.25833/7wqs-gb12 [122].

### 3.8. Generation of a Prostate Cancer-Related Gene List

In addition to identifying annotated pathogenic and likely pathogenic alterations as well as the top genes affected by SCNAs, we reviewed alterations involving potential PCa driver genes and non-coding events associated with prostate cancer. A list of 159 PCa-associated genes was compiled from recent studies that identified recurrently mutated genes in primary and metastatic samples (Appendix A) [6,7,8,9]. The list included commonly altered genes with potential functional relevance from The Cancer Genome Atlas (TCGA) primary PCa data [7,123], potential driver genes identified in primary and metastatic samples by Wedge et al. [8] and genes recurrently mutated in metastatic disease as identified by Robinson et al. [6] and Armenia et al. [9]. A list of non-coding events was compiled from recent published data (Table 4).

### 3.9. Other Analyses

The full list of binding clusters of 340 TFs compiled by the ENCODE project was obtained from the University of California Santa Cruz data repository (encRegTfbsClistered table; last updated 16 May 2019) and examined for somatic variants using a custom R script. Somatic variants within AR binding sites were evaluated against published putative binding sites observed in the LNCaP prostate cancer cell line (NCBI Gene Expression Omnibus accession GSE83860; [33]). The Circos plots in Figure 3, Figure 4, Figure 5, Figure 6 and Figure 7 were generated using the CIRCOS software (v0.69-6) [124] based on SNV/indel data from MUTECT, copy number data from CNVkit, and SV data from GRIDSS. Phylogenetic reconstruction of tumour evolution for patient 19651 was performed using phyloWGS [116] based on SNV/indel data from MUTECT and copy number data from TITAN. Analyses of COSMIC Mutational Signature [125] clonal evolution was performed using the R package Palimpsest v1 [126] which utilized SNV data from MUTECT for estimates of mutation signature and SNV allele frequency data from MUTECT along with copy number segmentation data from Sequenza [106] for estimates of clonality. 

## 4. Discussion

These real-world clinical cases demonstrate that clinically relevant mutations occur even in treatment-naïve patients across the spectrum of disease from high-risk primaries to metastatic cases. While the pathways impacted in these cases align with those identified in larger scale genomic studies, the coexistence of multiple alterations has not been explored. These findings raise several points.

Firstly, studies of neoadjuvant or adjuvant docetaxel in men with high-risk localized disease undertaking radical prostatectomy or definitive radiotherapy with ADT have had conflicting results [4,46,71] but a subgroup of men appear to benefit. Poor prognostic genomic findings, such as *TP53* deletions or deleterious *BRCA2* alterations at baseline may be useful in selecting men for additional treatment. Similarly, not all men require escalated treatment beyond ADT for HSPC but biomarkers to guide treatment selection remain limited. The findings of *TP53* and/or Wnt pathway activating alterations in 5/8 (63%) of our primary samples highlight a potential biomarker for selecting men that should be considered for escalated therapy, preferably with docetaxel rather than a novel ASI [32,62,127]. Though speculative in the hormone sensitive setting, there is mounting evidence these alterations could be useful in guiding treatment selection in CRPC. Secondly, we observed events in minimally treated patients, such as *NCOR1* and *NCOR2* losses, that may be associated with ADT resistance. These alterations again may identify patients at risk for early development of CRPC who may need escalated therapy upfront. Thirdly, pathway mutations typically enriched in metastatic CRPC, particularly PI3K and MAPK pathway SCNAs, were frequently seen in our patients and represent potential targets for neoadjuvant intervention in high risk localized and/or de novo metastatic HSPC clinical trials. 

The addition of WGM in our study did not identify a current therapeutic target but it did identify SVs impacting oncogenic and tumour suppressing genes that were not identified by using WGS alone. Though we did identify non-coding events affecting the promoters, enhancers and TF binding sites of relevant genes, their therapeutic relevance has yet to be elucidated. However, as WGS and WGM data accumulate and annotations improve, we may find new relevant mutations and begin to understand how they may be integrated into clinical practice. Additionally, the use of complementary genomic technologies such as RNA-sequencing and chromatin immunoprecipitation sequencing may improve our ability to translate genomic data into real-world clinical decision-making. 

In this retrospective study, we assess the current status of genome profiling, specifically WGS and WGM, to inform decision-making for 13 patients presenting with metastatic PCa. Our findings suggest that, despite being a cancer associated with a low TMB, individual PCas can harbour complex series of mutations affecting multiple growth pathways. Therefore, the precision medicine model of identifying one target to treat is unlikely to succeed. Given its heterogeneity and despite comprising only a very small fraction of the I-PREDICT study cohort [128], PCa may be the ideal cancer to test the paradigm of using genomics to identify and treat multiple targets simultaneously. 

## 5. Conclusions

Our analyses demonstrate that whole genomic interrogation of PCas may provide invaluable information at any stage of the disease. Most of our cases had alterations affecting multiple signalling pathways highlighting the utility of a comprehensive molecular assessment in tailoring treatment strategies to an individual. Moreover, WGM identified SVs disrupting prostate cancer relevant genes that were not apparent on our WGS analyses. Many non-coding and WGM events were identified but their therapeutic relevance is yet to be established. Though these data add to our current knowledge, further research is needed, potentially integrating additional genomic technologies, to identify new treatment targets and predictive biomarkers. While several potential biomarkers that may influence treatment decisions were found in these patients, most have yet to be validated in prospective clinical trials. 

## Figures and Tables

**Figure 1 cancers-12-01178-f001:**
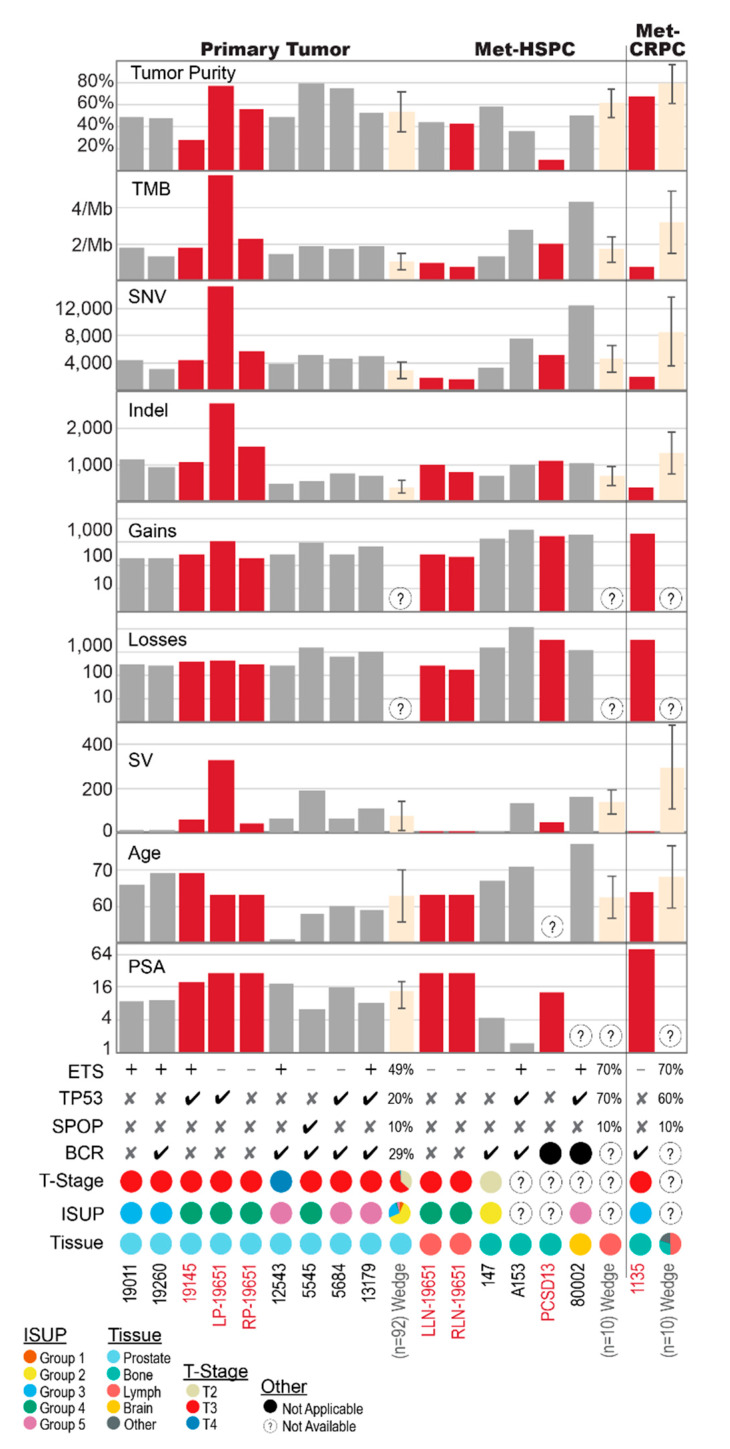
Summary of genomic landscape relevant to tumour purity and related to clinical and pathological features for 16 samples from 13 patients, and further compared to the Wedge et al. data. Met-HSPC: metastatic hormone sensitive PCa; met-CRPC: metastatic castration-resistant PCa; TMB: Tumour mutational burden; SNV: single nucleotide variants; Indel: small insertion or deletion; Gains and Losses: somatic copy number alterations (SCNA); SV: Structural variation including large insertions or deletions, inversions, translocations and duplications; PSA: prostate-specific antigen (ug/L); ETS: presence of ETS fusion event; TP53: presence of *TP53* alteration including SNV, SCNA or SV; SPOP: presence of *SPOP* SNV; BCR: biochemical recurrence; ISUP: International Society of Urological Pathologists cancer grade (correlates to Gleason scores). Error bars for Wedge et al. data reflect +/− one standard deviation of the sub-group totals. Sample identifiers in red, with matching red bar plots, are indicative of patients pre-treated with ADT.

**Figure 2 cancers-12-01178-f002:**
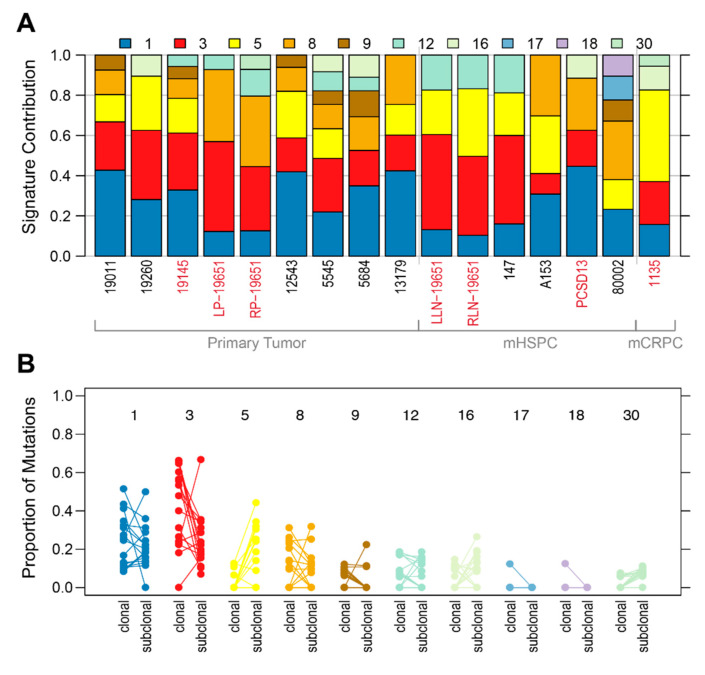
COSMIC Mutational Signatures (**A**) Proportion of signatures in each sample, for Signatures with >5% contribution; (**B**) Clonal vs. subclonal signature exposures. mHSPC: metastatic hormone-sensitive prostate cancer; mCRPC: metastatic castration-resistant prostate cancer.

**Figure 3 cancers-12-01178-f003:**
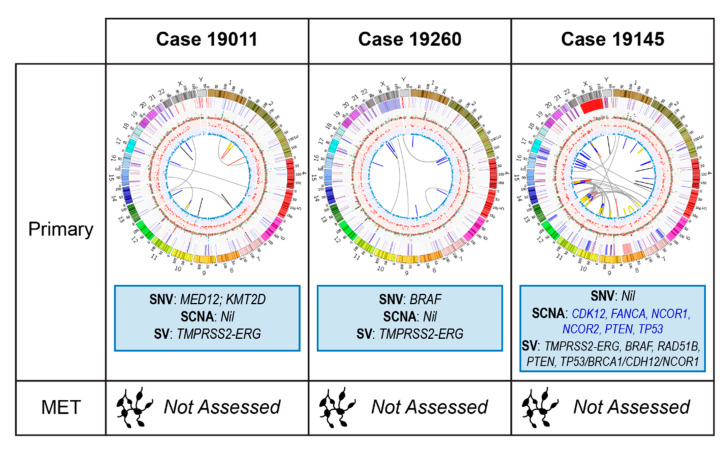
Summary of genomic alterations in primary prostate samples with synchronous lymph node metastases (Cases 19011, 19260 and 19145). Relevant somatic variants by type listed with SCNAs: blue indicates loss. Circos plots depict mutational load in each tumour sample. The outermost (first) track: autosome (chromosomes 1 to Y) ideograms with centromeres shown in red and the pter-qter orientation in a clockwise direction (length in Mbp); second track: somatic copy-number gains (red) and losses (blue); third track: somatic SNV allele frequencies (not corrected for tumour purity) coloured according to their mutation changes per Alexandrov et al. [39]; fourth and fifth tracks: allele frequencies (not corrected for tumour purity) of small deletions (red) and insertions (blue); innermost circle: acquired genomic rearrangements, including deletions (blue), tandem duplications (red), inversions (orange), insertions (black) and interchromosomal translocations (grey). MET: metastases.

**Figure 4 cancers-12-01178-f004:**
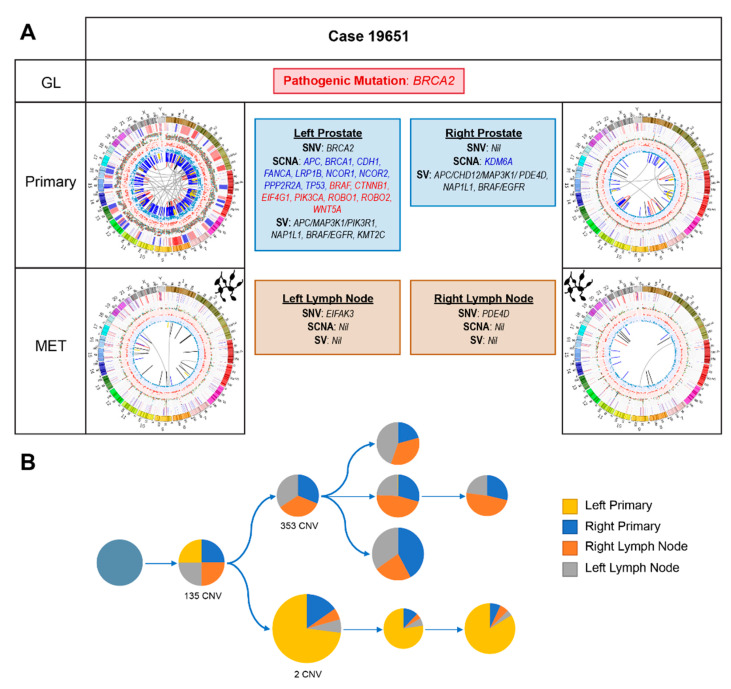
Case 19651. (**A**) Summary of relevant genomic alterations (germline and somatic); Circos plots as per Figure 3. GL: germline mutations; MET: metastases; (**B**) Phylogenetic reconstruction of cancer evolution predicted based on somatic SNV and copy number data. Each circle is a predicted tumour subclone, from the leftmost ancestral clone, with pie charts representing cancer cell fractions (proportion of the four samples harbouring the corresponding clone). Sizes of the circles are proportionate to the number of additional small somatic mutations acquired. Number of SCNAs acquired are indicated.

**Figure 5 cancers-12-01178-f005:**
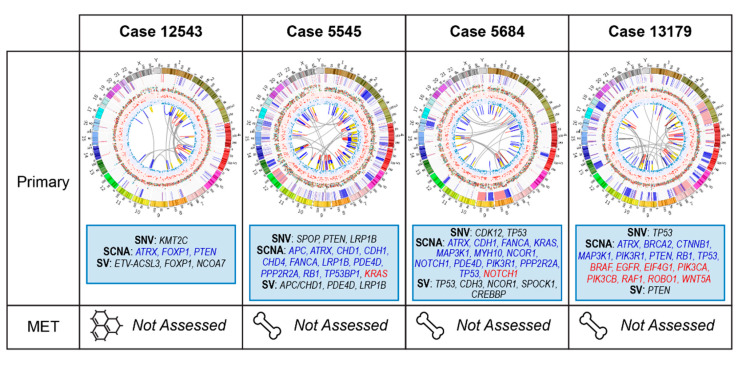
Summary of relevant genomic alterations for cases with subsequent metastatic relapse post radical prostatectomy (cases 12543, 5545, 5684, 13179); Circos plots as per Figure 3, with red text indicating SCNA loss.

**Figure 6 cancers-12-01178-f006:**
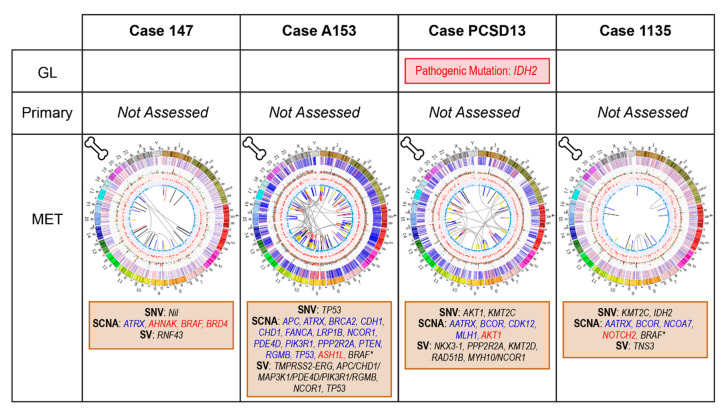
Summary of relevant genomic alterations for cases with metastatic disease at the time of sampling (cases 147, A153, PCSD13, 1135); Circos plots as per Figure 3.

**Figure 7 cancers-12-01178-f007:**
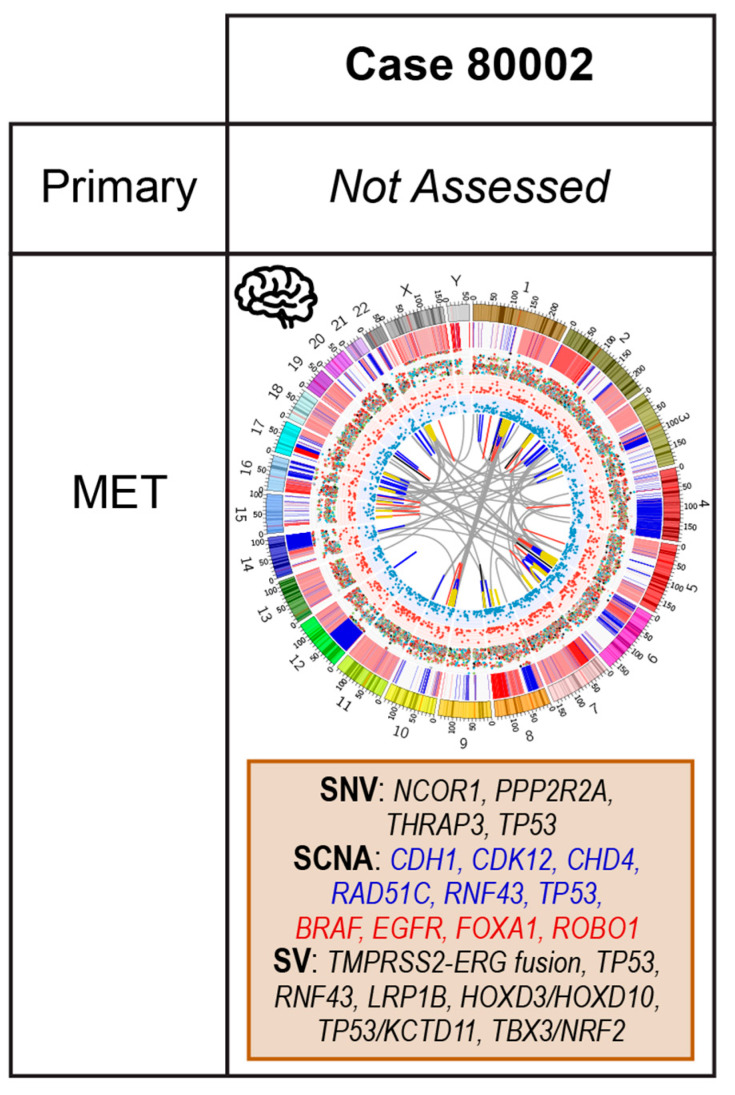
Summary of relevant genomic alterations for case 80002 with a brain metastasis at the time of sampling (80002) and Circos plots as per Figure 3.

**Table 1 cancers-12-01178-t001:** Clinical and pathological characteristics.

Diagnosis					Time of Sample Collection for Genomic Interrogation							Relapse and Outcomes			
Patient ID	Age	Stage	Initial Treatment	ISUP	Clinical State	PSA	ECOG	Symptoms	Sample Site	ADT Prior	CRPC?	Time to BCR (mos)	Rx for BCR	Time to Mets (mos)	Duration Follow-Up (mos)
1135	64	T3N0	RP	3	Rl	80.7	1	Yes	Bone	Yes	Yes	33	ADT	68	68
19651	63	T3N1	ADT, RP, aRT	4	Dx	28	0	No	Prostate, nodes	Yes (6wks)	No	NR	NA	NR	20
147	67	T2N0	RP	2	Rl	4.4	0	Yes	Bone	No	No	84	Nil	107	135
19260	69	T3N1	RP	3	Dx	9.2	0	No	Prostate	No	No	16	sRT	NR	27
5545	58	T3N0	RP	4	Dx	6.3	0	No	Prostate	No	No	6	sRT	24	72
5684	60	T3N0	RP	5	Dx	15.7	0	No	Prostate	No	No	36	ADT	120	132
19145	69	T3N1	RP, aRT	4	Dx	20	0	No	Prostate	Yes (4wks)	No	NR	NR	NA	33
19011	66	T3N1	RP, aRT	5	Dx	8.9	0	No	Prostate	No	No	NR	NA	NA	35
12543	51	T4N0	RP, aRT	5	Dx	18.6	0	No	Prostate	No	No	72	ADT	NA	120
13179	59	T3N0	RP, aRT, ADT	5	Dx	8.4	0	No	Prostate	No	No	51	ADT	51	51
PCSD13	69	TxNxM1	ADT	-	Dx	12.8	2	Yes	Bone	Yes (8wks)	No	NA	NA	NA	9
A153	71	T2N0	RP	3	Rl	1.5	0	Yes	Bone	No	No	93	Nil	105	120
80002	77	TxNxM1	Resection, ADT	5	Dx	-	1	Yes	Brain	No	No	NA	NA	NA	1

Dx = Diagnosis; Sample = time of tissue biopsy for genomic interrogation; ADT = Androgen deprivation therapy; RP = radical prostatectomy; aRT = adjuvant radiotherapy; sRT = salvage radiotherapy; Rl = Relapse; ECOG = Eastern Cooperative Oncology Group Performance Status; wks: weeks; BCR = Biochemical relapse; mos = months; NR = Not relapsed; NA = Not applicable.

**Table 2 cancers-12-01178-t002:** Genomic alterations affecting key genes in the PI3K, MAPK and WNT signaling pathways.

PI3K Pathway			MAPK Pathway			WNT Pathway		
Gene	Cases	Event	Gene	Cases	Event	Gene	Cases	Event
*PTEN*	554512543, 1914519145	SNVSCNASV	*BRAF*	1926019651LP, 13179, 1135, 147, A153, 8000219145,19651LP+RP, 80002	SNVSCNASV	*APC*	19651LP, 5545, A15319651LP+RP, 12543, 5545, A153	SCNASV
*PIK3CA*	19651LP, 13179	SCNA	*EGFR*	13179, 80002	SCNA	*CTNNB1*	19651LP, 13179	SCNA
*PIK3CB*	13179	SCNA	*KRAS*	19145, 5545, 5684	SCNA	*RNF43*	80002, 14780002, 147	SCNASV
*PIK3R1*	5684, 13179	SCNA	*MAP3K1*	5684, 13179,A153, 19651RP+LP	SCNASV	*WNT5A*	19651LP, 13179, 8000219260	SCNASV
*AKT1*	PCSD13	SNV, SCNA	*RAF1*	19651LP, 13179, 147, 80002	SCNA	*MED12*	19011	Germline SNV

**Table 3 cancers-12-01178-t003:** Structural variants identified by optical mapping as compared to whole genome sequencing. INS: Insertion; DEL: deletion; DUP: duplication; INV: inversion; Intra-Chr: intrachromosomal translocation; Inter-Chr: interchromosomal translocation.

Disease State of Sample	Sample ID	SVs from Whole Genome Optical Mapping							% Missed by Whole Genome Sequencing						
		Total	INS	DEL	DUP	INV	Intra-Chr	Inter-Chr	Total	INS	DEL	DUP	INV	Intra-Chr	Inter-Chr
Primary Tumor	5545	39	0	27	3	0	7	2	36	-	37	67	-	29	0
	13179	2	0	2	0	0	0	0	100	-	100	-	-	-	-
	5684	5	3	2	0	0	0	0	100	100	100	-	-	-	-
	12543	10	1	5	0	0	1	3	60	0	40	-	-	100	67
	19651LP	91	5	33	3	1	19	30	29	60	39	0	0	11	27
Met HSPC	A153	8	6	1	1	0	0	0	100	100	100	100	-	-	-
	147	5	4	0	1	0	0	0	100	100	-	100	-	-	-
	19651LLN	10	0	10	0	0	0	0	100	-	100	-	-	-	-
	80002	70	2	25	13	1	9	20	54	50	52	100	100	56	25
MetCRPC	1135	6	2	4	0	0	0	0	100	100	100	-	-	-	-

**Table 4 cancers-12-01178-t004:** Sites of recurrent non-coding alterations reported in the literature with potential clinical relevance.

Nearby Gene	Variant Positions	Data from Literature	Reference	Patient IDs
*NEAT1*	Chr11:65,190,268-65,213,009	13/112 cases, 6/20 mets all with previous ADT. *NEAT1* produces a long non-coding RNA that regulates several growth pathways and overexpression is associated with PCa progression	Wedge [8]	5545
*FOXA1*	Promoter, Chr14:37587200-37597201	14 coding and 6 non-coding mutations; regulates AR signalling	Wedge	5684
*FOXA1*	Chr14:37886261-37888565, 37903630-37906634, 38035667-38036817, 38053354-38056060, 38056084-38059097, 38127358-38128083	*FOXA1* is a co-factor for AR. These are *cis-*regulatory elements	Zhou [34]	5684
*AR*	Upstream promoter	Tandem duplications, 70–87% mCRPC vs. <2% primary PCa	Viswanathan [11]	Nil
*AR*	ChrX: 66117800-66128800 (66.10–66.20 bin)	I peak, long range enhancer of AR, only 1/54 primary samples (Viswanathan); Copy number gain results in proliferation in low androgen condition and enzalutamide resistance	Takeda [35], Viswanathan	Nil
*AR*	Transcription Factor Binding Sites	Recurrently altered in primary PCa	Morova [33]	1135, 5545, 5684, 12543, 13179, 19011, 19145, 19260, 80002, 19651 (LP, RP, RLN), A153, PCSD13
*MYC*	Chr8: 128.14–128.28, 128.47–128.54, 128.54–128.62	8q24 risk loci PCa, associated with MYC enhancer activity	Ahmadiyeh [37], Yeager [36]	19651LP, 12543, A153

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
