# Peer review of "The Impact of Whole Genome Data on Therapeutic Decision-Making in Metastatic Prostate Cancer: A Retrospective Analysis"

_cancers, 2020, doi:10.3390/cancers12051178_

Round 1

Reviewer 1 Report

The manuscript by Crumbaker, et al. aims to study the therapeutic impact on metastatic prostate cancer by interrogating genomic events through whole genome sequencing on a small number of patients. Here are some of my comments/questions related to the manuscript.

  1. Does the number of structural variant (SV), whether relatively higher or lower provide a better understanding on the genomic instability among the group of patients considered in the study in comparison to the patien's treatment? Could it be used as a factor on deciding treatment response?
  2. As the authors have dived deep into the 9 patient subsets, what else was detected or identified beyond the SVs, especially if there was any large complex genomic rearrangements or chained fusions? If yes, does these have any impact on the treatment respond?
  3. Does the authors mean by interchromosomal translocation as chromothripsis or do they indicate these as a neochromosomal fusion? Further explanation is required.

Author Response

  1. Does the number of structural variant (SV), whether relatively higher or lower provide a better understanding on the genomic instability among the group of patients considered in the study in comparison to the patient's treatment? Could it be used as a factor on deciding treatment response?

Response: Thank you for this interesting question. Unlike tumor mutational burden (TMB) which is based on the number of SNVs and InDels, at present, it is unknown whether the number of SVs correlate to treatment response. To the best of our knowledge, this is the first comprehensive look at SVs in metastatic hormone sensitive disease, but the numbers are too small to draw any conclusions on treatment impact. While TMB and number of SVs correlated in many of our cases, in some they did not (1135, 5545, 13179, 19011, 19260).

  1. As the authors have dived deep into the 9 patient subsets, what else was detected or identified beyond the SVs, especially if there was any large complex genomic rearrangements or chained fusions? If yes, does these have any impact on the treatment respond?

Response: Though we did identify some large SVs with our whole genome mapping approach, there was no evidence to suggest that more complex rearrangements were present, as would be indicated by local clusters of SVs and/or excessive local amplifications. Despite there being many SCNAs (please see Figure 1), the overall number of SVs was low.

  1. Does the authors mean by interchromosomal translocation as chromothripsis or do they indicate these as a neochromosomal fusion? Further explanation is required.

Response: Interchromosomal translocations are defined as fusion junctions between two pieces of DNA from different chromosomes that are not meant to be joined. We did not perform additional analyses, such as chromosomal painting, to determine whether the interchromosomal translocations were due to neochromosomal fusion, but the low number of SVs in the samples makes it unlikely that neochromosomes or chromothripsis were present.

Reviewer 2 Report

  1. Discussion should come after Materials and Methods
  2. The manuscript is too long and verbose
  3. The patients should be characterized by +/- symptoms, +/- metastases, +/- disease progression on chemical castration, performance status
  4. Which AUA/NCCN guideline therapy were each patient receiving while undergoing genomic analysis?
  5. The presence of actionable genomic pathways in advanced oncologic cases is not novel. This study would be novel if the authors showed a survival benefit after administering the therapy corresponding to the pathway relevant for each patient.

Author Response

  1. Discussion should come after Materials and Methods.

Response: Discussion has been moved as requested, however we note that the word template and instructions for authors provided by the journal request that the Methods come after the Discussion.

  1. The manuscript is too long and verbose.

Response: We have shorted the paper from 8400 words to 7500 words, while summarizing much of the clinical and pathological data in the new Table 1.

  1. The patients should be characterized by +/- symptoms, +/- metastases, +/- disease progression on chemical castration, performance status.

Response: Although patient characteristics are mentioned for each patient in a case-by-case evaluation, we have expanded Table S1 and moved it to the main text, now Table 1.

  1. Which AUA/NCCN guideline therapy were each patient receiving while undergoing genomic analysis?

Response: As per point 3 above, treatment received prior to genomic analysis has been summarized, for convenience, in new Table 1.

  1. The presence of actionable genomic pathways in advanced oncologic cases is not novel. This study would be novel if the authors showed a survival benefit after administering the therapy corresponding to the pathway relevant for each patient.

Response: We appreciate the benefit for establishing the survival impact of precision medicine-based therapies but this was not possible in the scope of this study. While we agree that the identification of actionable genomic pathways in advanced, particularly castrate resistant, prostate cancer cases is not novel, our approach differs to that of previously published studies.

  • Our cohort consists of predominantly hormone-sensitive locally advanced and metastatic tumor samples, which are under-represented in the current literature that primarily consists of moderate-risk, localized tumors and heavily pre-treated castrate-resistant metastases
  • Additionally, these previous studies are population based where mutations are described as a cohort with little consideration to their interactions and the context at an individual level

Therefore, the novelty of our approach is in our assessment of the applicability of genomic information in an individual, case-by-case analysis, i.e. a true precision medicine model. Despite prior studies identifying high rates of “clinically actionable mutations”, as an oncologist and first author, I understand that the reality is that few of the targets identified have established therapeutic relevance in prostate cancer and it is important to continue to add to the knowledge base while contextualizing new and existing findings.

Reviewer 3 Report

This manuscript is  evaluating the genomic landscape of 33 metastatic prostate cancer, utilizing this information to inform personalized treatment is in its
34 infancy. They performed a retrospective pilot study to assess the current impact of precision medicine 35 for locally advanced and metastatic prostate adenocarcinoma and evaluate how genomic data could 36 be harnessed to individualize treatment.This retrospective oncological assessment highlights the genomic complexity of prostate cancer and the potential impact of assessing genomic data for an individual at any stage of the disease.The content innovation, but there are the following questions need to be addressed.

  1. The experimental method needs to be clarified. How to obtain tumor samples, whether the tumor has pathological confirmation, and whether the location of the tumor has been determined?
  2. Please provide the pathological section report of the tumor, and the corresponding to the genome figure.
  3. Are all of these MRCP patients operation before?
  4. Please provided the IHE of the tumor marker?
  5. This study will be more reliable if it is combined with real time-RT-PCR study report.

Author Response

  1. The experimental method needs to be clarified. How to obtain tumor samples, whether the tumor has pathological confirmation, and whether the location of the tumor has been determined?

Response: Thank you for raising this issues, we have added clarification to the methods section at lines 519-524 (see below).

Primary tumor samples were collected at the time of radical prostatectomy and two core biopsies were taken from the prostate regions with cancer on pre-operative biopsy. Lymph node tissue was collected at time of radical prostatectomy from nodal masses with palpable tumor. Metastatic samples were obtained by image guided biopsy or at surgical resection (80002, PCSD13). All tissue samples were snap frozen. The presence of prostate cancer and its location witin the samples was confirmed by a pathologist prior to dissection for DNA extraction.

  1. Please provide the pathological section report of the tumor, and the corresponding to the genome figure.

Response: We have added further pathological information to the main text in the newly made Table 1, in addition to the information previously provided in Figure 1. However, given the space limitations, usual practices and the focus of the paper, it was not feasible to add additional pathological information to the figures. We are happy to provide these reports upon request.

  1. Are all of these MRCP patients operation before?

Response: All but case 1135 were hormone sensitive at the time of sampling/surgery. We have added a column to Table 1 to further clarify their CRPC status and Figure 1 already highlights those that had ADT prior to sampling (though only 1 had developed CRPC).

  1. Please provided the IHE of the tumor marker?

Response: All of these patients had standard IHC findings for prostate adenocarcinoma. As such we have not provided their IHC profiles except for case 80002. In the case of 80002, given the unusual location of the metastasis (brain), we had highlighted that their morphology and IHC findings were consistent with prostate adenocarcinoma. Due to the number of variants identified, we were unable to perform additional IHC to confirm their impact.

  1. This study will be more reliable if it is combined with real time-RT-PCR study report.

Response: We agree that for prospective therapeutic decision-marking, RT-PCR would be desirable for certain alterations and will need to be integrated into future precision medicine pipelines. However, for research purposes and the purposes of our study, because co-isolation of RNA with high molecular weight DNA is not possible and co-isolation with standard DNA reduces the quality of the DNA, we opted not to extract RNA for RT-PCR.